⊘ | **Open Peer Review** | Computational Biology | Research Article

# Group I introns in tRNA genes of Patescibacteria

Yuna Nakagawa,[1,2] Kazuaki Amikura,[2] Kimiho Omae,[2] Shino Suzuki[2,3]

**ABSTRACT**  Introns are generally considered rare in bacteria, yet they are frequently observed in Patescibacteria, which have highly reduced genomes. To systematically explore the diversity, roles, and evolution of introns in Patescibacteria, we first focused on the tRNA introns. Using 95 complete genomes, we identified $tRNA^{Asn}$ and $tRNA^{Asp}$ genes previously undetected by standard annotation tools due to group I introns inserted at an unusual position, 35/36, in the anticodon loop. *In vitro* splicing assays confirmed that these introns catalyze precise self-splicing, validating our computational approach. A large-scale survey of complete bacterial genomes revealed that intron insertions at position 35/36 are highly enriched in Patescibacteria but rare in other phyla. Subgroup classification indicated that 81% of all tRNA introns belong to the IC subgroup, whereas nearly all Patescibacteria introns were classified as IA. As most tRNA introns lack homing endonuclease genes, horizontal transfer appears limited. Comparative analysis across bacterial phyla showed that Patescibacteria and Cyanobacteriota exhibit the highest prevalence of group I introns (~40% of genomes). In contrast, group II introns, which require protein cofactors for activity, were more common in other bacteria, including Cyanobacteriota, but absent in Patescibacteria. Collectively, these findings suggest that Patescibacteria harbor introns with phylum-specific trends in abundance, structure, and evolutionary lineage. The coexistence of extensive genome reduction and persistent group I introns may reflect an adaptive strategy, where introns serve as efficient RNA-based regulatory elements, potentially substituting for complex protein-mediated systems.

**IMPORTANCE** Introns were traditionally thought to be rare in bacteria, yet their occurrence and diversity may have been underestimated. Here, we present the first comprehensive overview of group I and group II introns in Patescibacteria. While most introns are readily identified, group I introns inserted at position 35/36 within the anticodon loop often escape detection by standard annotation tools; through experimental verification, we demonstrate that these introns are accurately spliced despite their unusual insertion site. Notably, approximately 40% of genomes in both Patescibacteria and Cyanobacteriota harbor group I introns; however, while around 20% of Cyanobacteriota genomes also contain group II introns, none were detected in Patescibacteria. These results illustrate a previously overlooked phylogenetic distribution of group I and group II introns across the bacterial domain.

**KEYWORDS**  Patescibacteria, group I intron, group II intron, genome analysis, tRNA, tRNA intron

Recent advances in genome-resolved metagenomics have uncovered numerous previously unrecognized microbial lineages, fundamentally reshaping our understanding of microbial diversity and evolution. Among these, the phylum Patescibacteria, previously referred to as the Candidate Phyla Radiation (CPR), has attracted particular attention due to its remarkable diversity (1), uncertain phylogenetic position within the bacterial domain (2), and unusual biological features (3). Patescibacteria is a diverse

**Peer Reviewer** Jakub Barylski, Uniwersytet im Adama Mickiewicza w Poznaniu Wydzial Biologii, Poznan, Poland

Address correspondence to Shino Suzuki, shino.suzuki@riken.jp.

The authors declare no conflict of interest.

See the funding table on p. 18.

clade of largely uncultivated organisms that are broadly distributed across environments, including groundwater aquifers (4), soils (5), freshwater systems (6), and oral microbiomes (7). They are characterized by ultra-small cell sizes (approximately 0.2 µm–0.3 µm) and highly reduced genomes ranging from 0.5 to 1.5 Mbp (8–10). Genomic analyses indicate that these organisms lack essential biosynthetic pathways and rely on interactions with other microbes for survival. Consistent with this, the few members that have been cultivated are epibionts attached to the host cell surfaces (7, 11–14). In addition to their streamlined metabolism, Patescibacteria display unusual ribosomal features, characterized by the loss of several ribosomal proteins (3, 15) and reduced sets of ribosome biogenesis factors (16). All these unusual characteristics position Patescibacteria as a reservoir of unexplored biological knowledge, with profound implications for microbial physiology, evolution, and translational mechanisms.

Although many aspects of Patescibacteria remain enigmatic, one intriguing feature is the widespread occurrence of introns within their compact genomes, despite the general rarity of introns in bacteria. Introns are defined as nucleotide sequences that are removed from precursor RNAs through splicing, resulting in the formation of mature mRNAs or functional RNAs. Four major types of introns are recognized: group I and group II introns (17), bulge-helix-bulge (BHB) motif introns (18), and spliceosomal introns (19). In bacteria, only group I and II introns have been reported, whereas BHB motif and spliceosomal introns are characteristic of Archaea and Eukarya, respectively. Introns are not merely noncoding sequences, but they play diverse regulatory and physiological roles, including the control of mRNA expression (20), the regulation of tRNA maturation (21), and the promotion of gene rearrangement and genome plasticity (22).

Group I and II introns, the only types of introns identified in bacteria, are self-splicing ribozymes, catalytically active RNA elements capable of excising themselves from precursor transcripts. The two types differ in their RNA secondary structures and splicing mechanisms: group I introns utilize an exogenous guanosine to mediate cleavage and ligation without the need for protein cofactors (23, 24), whereas group II introns form a lariat structure and generally require intron-encoded protein cofactors for efficient splicing (25, 26). In bacteria, allosteric group I introns have been discovered, suggesting that bacterial introns are not necessarily selfish genetic elements, but may also play regulatory or functional roles (27).

In most bacterial species, rRNA and tRNA genes are considered to be intron-less, with only rare exceptions (3, 10, 28–32). However, members of the phylum Patescibacteria, which belong to the domain Bacteria, frequently harbor introns within rRNA genes (3, 33) and, more rarely, within tRNA genes (10). Group I and II introns are known to possess mobility, which may further contribute to genome dynamics through intron gain and loss events over evolutionary timescales (34–37). However, the diversity and function of tRNA introns in Patescibacteria remain poorly understood.

Here, to gain insight into the diversity of intron-containing tRNAs, we systematically analyzed introns in bacterial tRNAs. Given that tRNA and rRNA genes are often poorly assembled and/or mis-binned in metagenome-assembled genomes, we used complete genomes from the Genome Taxonomy Database (GTDB) release 220 (38). Our analysis revealed a diversity of introns in tRNAs. Notably, some tRNA$^{Asn}$ and tRNA$^{Asp}$ genes initially appeared to be absent but were actually present and disrupted by introns inserted at uncommon sites within the anticodon loop. *In vitro* assays confirmed that these introns are catalytically active and capable of producing mature tRNAs via self-splicing. Comprehensive genomic analysis of introns in tRNAs demonstrates that even the highly reduced genomes of Patescibacteria can retain functional self-splicing tRNA introns, which may play important roles in gene regulation and genome evolution or persist as relics of ancient genomic architecture.

## RESULTS AND DISCUSSION

### tRNA$^{Asn}$ and tRNA$^{Asp}$ genes in Patescibacteria undetected by standard annotation tools

We analyzed 95 Patescibacteria genomes, all classified as "complete genomes" in GTDB r220 (Table S1), each comprising a single contig. To identify tRNA genes corresponding to all 20 canonical amino acids, we employed three publicly available annotation tools for tRNAs, including tRNAscan-SE 2.0 (39), ARAGORN (40), and tFind (41, 42). While tRNAscan-SE is optimized for detecting intron-less tRNA genes, its ability to identify intron-containing tRNAs is limited. ARAGORN is designed to detect not only canonical tRNA genes but also tmRNA genes, and it is capable of recognizing tRNAs with introns. tFind is a tool that was improved in the detection sensitivity of intron-containing tRNAs by integrating tRNAscan-SE, ARAGORN, and Infernal with a custom covariance model (41).

Despite employing three different annotation tools on complete genomes, 10 of the 95 Patescibacteria genomes lacked one of the canonical 20 amino acid tRNAs (Table S2). Absence of tRNA$^{Asn}$ and tRNA$^{Asp}$ was striking (Fig. 1a), and no tRNA$^{Asn}$ genes were detected in at least 7 of the 95 Patescibacteria genomes and no tRNA$^{Asp}$ genes in two genomes. These unexpected absences were distributed across multiple classes within the phylum Patescibacteria, suggesting that currently available annotation tools remain insufficient for the reliable identification of certain tRNA genes in this phylum.

### Group I intron insertions at the position 35/36 in tRNA$^{Asn}$ and tRNA$^{Asp}$

Given the apparent absence of certain tRNA genes on the complete genomes (Fig. 1a), we performed additional searches to identify the corresponding tRNA genes. First, we detected candidate introns in the 95 complete genomes of Patescibacteria using Infernal with Rfam's RF00028.cm (a standard group I intron covariance model) and RF00029.cm (a standard group II intron covariance model). Intron-less tRNA genes, which were identified by tRNAscan-SE 2.0, were used as a database in BLAST searches, with intron-flanking sequences as queries, allowing identification of tRNA genes containing group I introns. The intron boundaries were consistently located in the anticodon loop, with a uridine at the end of the 5′ exon and a guanine at the 3′ end of the intron. The spliced tRNAs were predicted to adopt canonical secondary structures and to retain the anticodons of intron-less homologs (Fig. 1b; "Infernal-based") (see Materials and Methods).

This analysis predicted a total of 22 tRNA genes containing group I introns across 19 Patescibacteria genomes (Table 1; Table S3), whereas no group II introns were detected. Among these 22 tRNA genes, 15 were tRNA$^{Asn}$ (GUU) and 3 were tRNA$^{Asp}$ (GUC) genes. Notably, 10 of the tRNA$^{Asn}$ and 1 of the tRNA$^{Asp}$ genes were not detected by tFind, and these tRNA$^{Asn}$ and tRNA$^{Asp}$ genes harbor group I introns at position 35/36 (Fig. 2, Table 1), a site where no intron had been reported prior to the development of tFind (41). Further analysis revealed that ARAGORN and tFind annotated these intron-containing tRNA$^{Asn}$ genes as tRNA$^{Thr}$ (UGU) and the tRNA$^{Asp}$ genes as tRNA$^{Ala}$ (UGU), with introns incorrectly assigned to position 33/34 (Table 1). This discrepancy is likely due to a two-base shift in the predicted splicing boundary (Fig. 3). A previous study reported similarly that tFind occasionally annotated tRNA$^{Asn}$ (Asn.4Ug in the literature) as tRNA$^{Thr}$ (Thr.2Ug), because the latter received a slightly higher isoacceptor score (41). Comparable misannotation likely occurred for the tRNA$^{Asp}$ genes as well, illustrating the difficulty in consistently determining the correct splicing boundary between closely competing alternatives with existing annotation tools. Interestingly, we confirmed that tFind accurately detected tRNA$^{Thr}$ containing divergent group I introns in Patescibacteria (Table 1). Although these introns are comparable to canonical group I introns in overall length and predicted secondary structure, their sequences are highly divergent and therefore cannot be detected by either RF00028.cm (10) or ARAGORN (Table 1). The better performance of tFind is likely attributable to its use of the covariance model gpl_bact_tRNA.cm,

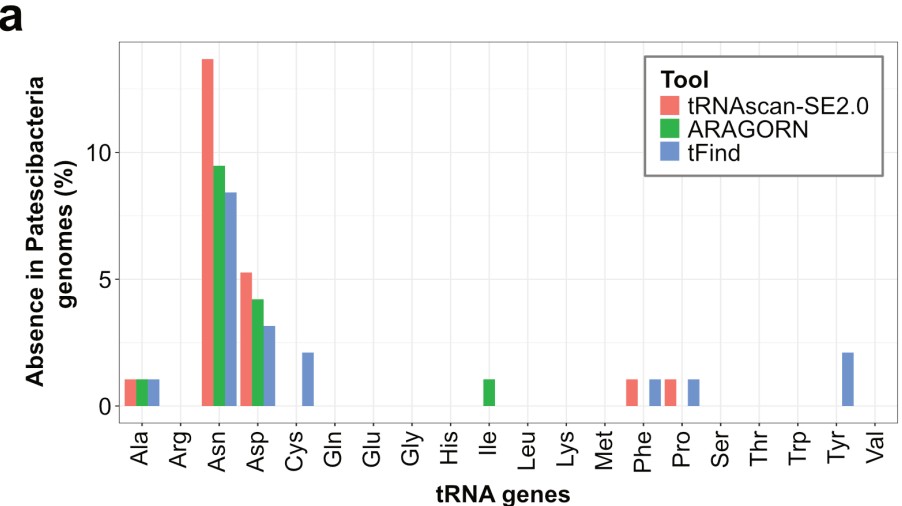

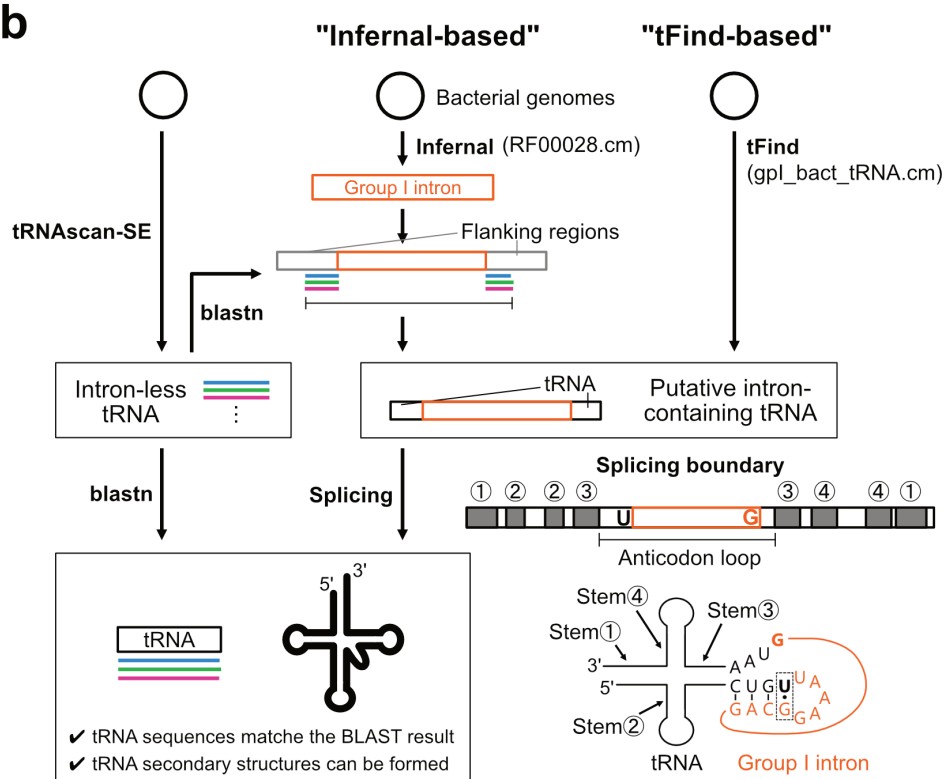

**FIG 1** Identification of group I introns inserted in tRNA genes. (a) Percentage of Patescibacteria genomes in which specific tRNA genes were not detected using tRNAscan-SE2.0 (red), ARAGORN (green), and tFind (blue). The analysis includes all 95 genomes of Patescibacteria registered as complete genomes in GTDB r220. The *x*-axis shows tRNA genes corresponding to the 20 canonical amino acids, and the *y*-axis shows the percentage of genomes lacking each tRNA gene. (b) Protocol for detecting tRNA genes with group I intron insertions in bacterial genomes. For Patescibacteria genomes, detection was validated using the "Infernal-based" protocol, whereas for all bacterial genomes, the "tFind-based" protocol was applied.

which was constructed from 26 representative group I intron sequences inserted into bacterial tRNA and tmRNA genes. To this end, we propose the following protocol (Fig. 1b; "tFind-based") as the most suitable for detecting amino acid tRNAs and their introns in Patescibacteria genomes: tFind is used to identify tRNAs. The sequences of tRNAs containing introns were compared with intron-less tRNA genes using BLAST to assess

**TABLE 1** tRNA genes with group I intron insertions detected in Patescibacteria genomes[a]

| Genome ID | This study | | ARAGORN | | tFind | |
|---|---|---|---|---|---|---|
| | tRNA | Position | tRNA | Position | tRNA | Position |
| GCA_000503835.1 | Asn (GUU) | 35/36 | Asn (GUU) | 35/36 | Asn (GUU) | 35/36 |
| GCA_000503915.1 | Asn (GUU) | 35/36 | Thr (UGU) | 33/34 | Thr (UGU) | 33/34 |
| GCA_001029735.1 | Asn (GUU) | 35/36 | Asn (GUU) | 35/36 | Thr (UGU) | 33/34 |
| GCA_016699245.1 | Asn (GUU) | 35/36 | Asn (GUU) | 35/36 | Thr (UGU) | 33/34 |
| GCA_016699425.1 | Asn (GUU) | 35/36 | Asn (GUU) | 35/36 | Asn (GUU) | 35/36 |
| GCA_016700015.1 | Asn (GUU) | 35/36 | Asp (GUC) | 37/38 | Asn (GUU) | 35/36 |
| GCA_016700035.1* | Asn (GUU) | 35/36 | Thr (UGU) | 33/34 | Thr (UGU) | 33/34 |
| GCA_016700135.1 | Asn (GUU) | 35/36 | Asn (GUU) | 35/36 | Asn (GUU) | 35/36 |
| GCA_016700315.1 | Asn (GUU) | 35/36 | Thr (UGU) | 33/34 | Thr (UGU) | 33/34 |
| GCA_016700375.1 | Asn (GUU) | 35/36 | Tyr (GUA) | 36/37 | Met (CAU) | 33/34 |
| GCA_023898525.1 | Asn (GUU) | 35/36 | Thr (UGU) | 33/34 | Thr (UGU) | 33/34 |
| GCA_025999335.1 | Asn (GUU) | 35/36 | Asn (GUU) | 35/36 | Thr (UGU) | 33/34 |
| GCA_026016225.1 | Asn (GUU) | 35/36 | Thr (UGU) | 33/34 | Thr (UGU) | 33/34 |
| GCA_030583765.1 | Asn (GUU) | 35/36 | Thr (UGU) | 33/34 | Asn (GUU) | 35/36 |
| GCA_030583845.1 | Asn (GUU) | 35/36 | Thr (UGU) | 33/34 | Thr (UGU) | 33/34 |
| GCA_001029755.1* | Asp (GUC) | 35/36 | Ala (UGC) | 33/34 | Asp (GUC) | 35/36 |
| GCA_023898525.1 | Asp (GUC) | 35/36 | Asp (GUC) | 35/36 | Ala (UGC) | 33/34 |
| GCA_030583725.1 | Asp (GUC) | 35/36 | Asp (GUC) | 35/36 | Asp (GUC) | 35/36 |
| GCA_001029755.1 | Met (CAU) | 33/34 | Met (CAU) | 33/34 | Met (CAU) | 33/34 |
| GCA_013426185.1 | Met (CAU) | 33/34 | Leu (CAA) | 36/37 | Met (CAU) | 33/34 |
| GCA_016699995.1 | Phe (GAA) | 33/34 | Phe (GAA) | 33/34 | Phe (GAA) | 33/34 |
| GCA_016699995.1 | Tyr (GUA) | 35/36 | Tyr (GUA) | 35/36 | Tyr (GUA) | 35/36 |
| GCA_001029635.1 | Not detected | | Arg (GCG) | 34/35 | Thr (GGU) | 36/37 |
| GCA_016700375.1 | Not detected | | Arg (GCG) | 34/35 | Thr (GGU) | 36/37 |
| GCA_030583745.1 | Not detected | | Thr (GGU) | 36/37 | Thr (GGU) | 36/37 |

[a]Detection results of tRNA genes with group I intron insertions (corresponding amino acids and insertion sites) in Patescibacteria genomes are compared among this study, ARAGORN, and tFind. Results from this study are based on the "Infernal-based" protocol shown in Fig. 1b and indicate that tRNA[Thr] introns at the 36/37 position were not detected using RF00028.cm, but were successfully identified by tFind using the gpl_bact_tRNA.cm. Rows marked with an asterisk (*) indicate the genome IDs and corresponding tRNA genes selected for splicing experiments in this study.

sequence similarity. In cases where no clear similarity was detected, the intron boundaries were refined based on the BLAST alignment results, and the corresponding tRNA sequences were reconstructed. Intron boundaries are consistently located within the anticodon loop, and the spliced tRNAs are predicted to fold into canonical secondary structures while retaining the expected anticodons.

## Experimental validation of splicing activity of group I introns inserted at position 35/36

Since tRNA[Asn] and tRNA[Asp] genes were predicted with group I introns at the new position of 35/36 in the anticodon loop, we conducted *in vitro* splicing assays to confirm the prediction by demonstrating functional self-splicing. We selected tRNA[Asn] and tRNA[Asp] gene sequences from the Patescibacteria genomes GCA_016700035.1 and GCA_001029755.1, respectively, as these represent the sole genomic copies of the respective tRNA genes interrupted at position 35/36 by a group I intron (Fig. 2; Table S4). Group I introns catalyze self-splicing through two transesterification reactions (Fig. 4a) (23, 24). In the first step, an exogenous guanosine (αG) binds to the active site and cleaves the 5′ splice site. In the second step, the terminal guanosine (ωG) replaces αG in the active site and mediates exon ligation at the 3′ splice site. During this process, the intron undergoes conformational changes that properly align the exons for catalysis. After ligation of the exons, the intron can circularize in the splicing assay (43, 44).

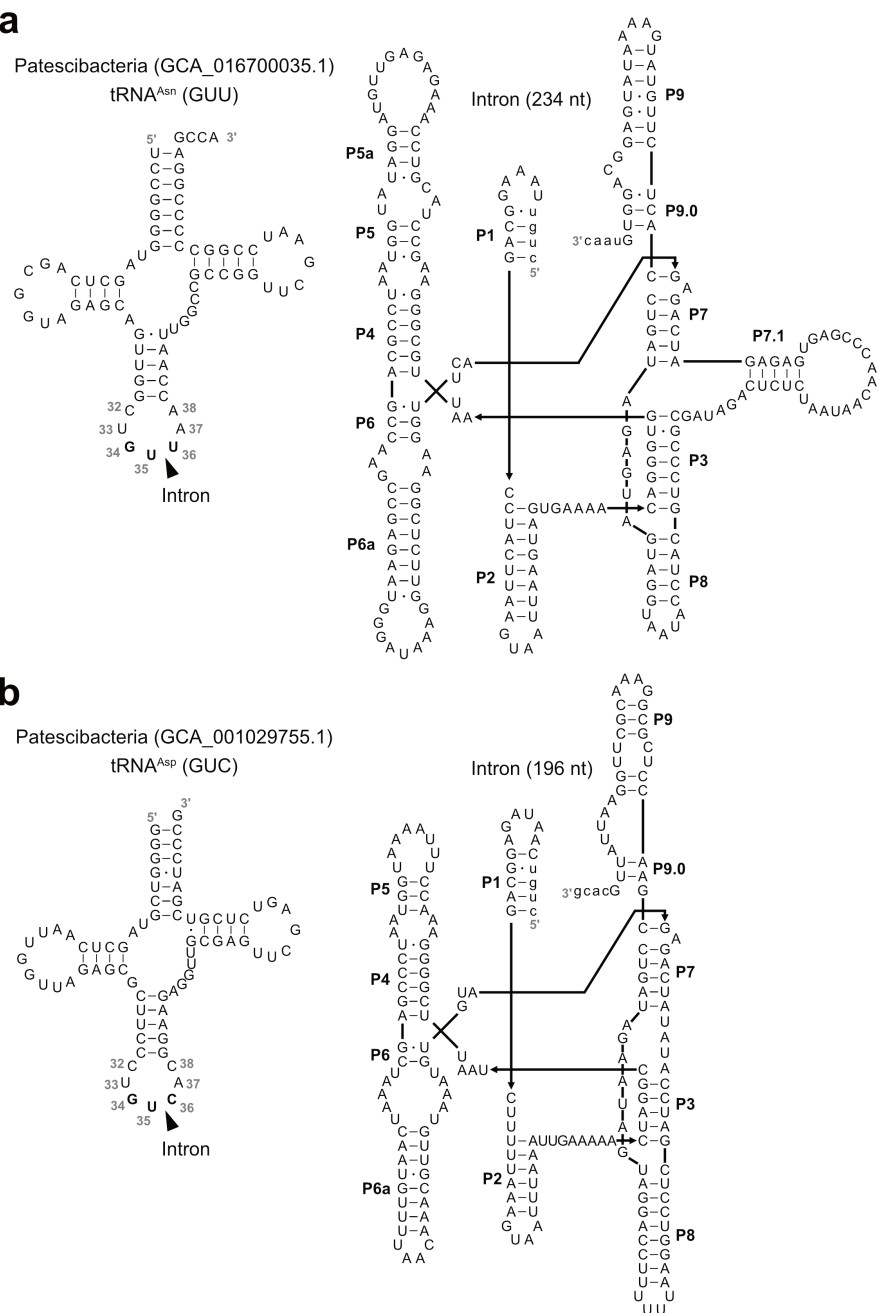

**FIG 2** Predicted secondary structures of tRNA genes and group I introns. (a) Predicted secondary structures of the tRNA[Asn] (left) and the group I intron inserted into it (right) from the Patescibacteria genome (accession: GCA_016700035.1). (b) Predicted secondary structures of the tRNA[Asp] (left) and the group I intron inserted into it (right) from the Patescibacteria genome (accession: GCA_001029755.1).

As a positive control, we used tRNA[Leu] (UAA) from Cyanobacteriota (*Nostoc* sp. PCC7120) that contains a group I intron and has previously been shown to splice efficiently under *in vitro* conditions (29). The assay produced the band pattern as expected, including the tRNA, linear intron (L-IVS), intermediate (I-IVS–3′ exon), and circular intron (C-IVS), thereby confirming the reliability of our experimental system (Fig. 4b). For the target sequences of tRNA[Asn] and tRNA[Asp] from Patescibacteria genomes, we detected distinct bands of spliced products as we observed in the positive control. The tRNA band was further sequenced and confirmed that the mature tRNAs

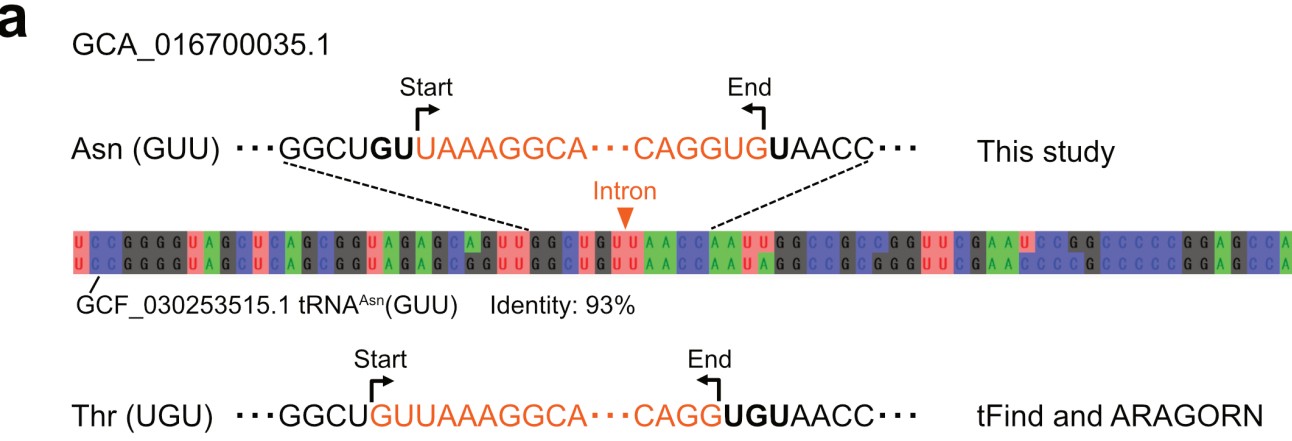

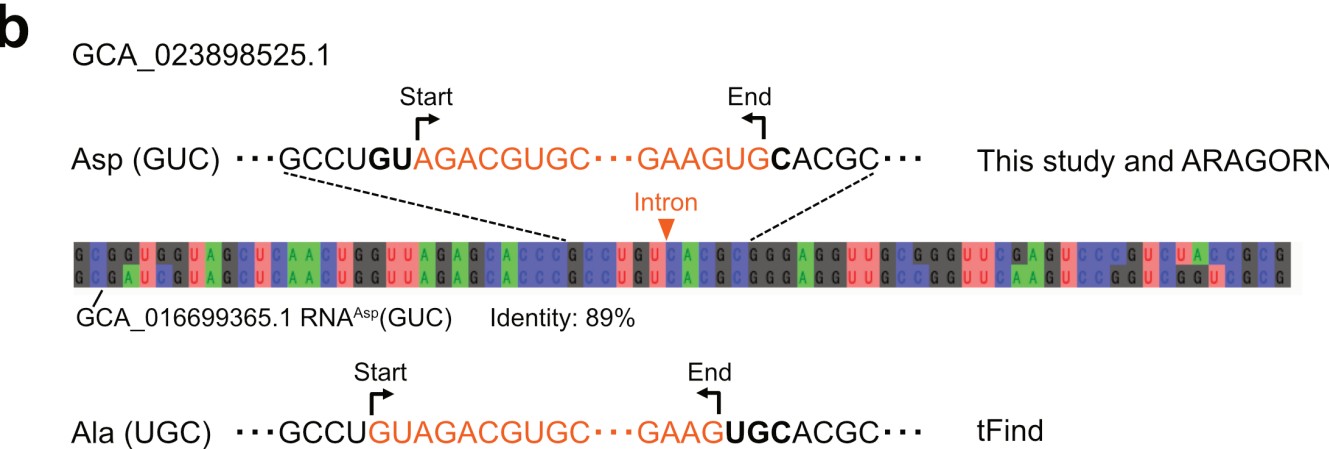

**FIG 3** Examples of cases where two possible tRNA gene boundaries can be defined. (a) In GCA_016700035.1, shifting the intron boundary by two nucleotides changes the prediction from tRNA[Asn] (GUU) to tRNA[Thr] (UGU). Both ARAGORN and tFind predicted tRNA[Thr], but our "Infernal-based" protocol predicted tRNA[Asn], and the sequence identity to tRNA[Asn] from GCF_030253515.1 was 93% (best hit). (b) In GCA_023898525.1, shifting the intron boundary by two nucleotides changes the prediction from tRNA[Asn] (GUU) to tRNA[Ala] (UGC). Our "Infernal-based" protocol and ARAGORN predicted tRNA[Asn], whereas tFind predicted tRNA[Ala]. The best BLAST hit was to tRNA[Asn] from GCA_016699365.1, with 89% (best hit).

contained the expected sequences, exhibiting precise intron excision at position 35/36 within the anticodon loop (Fig. 4c and d). As a negative control, we synthesized DNA templates in which the 3′ terminal G, the recognition site for splicing, was replaced with C and transcribed them into RNA for the assay. These RNA transcripts failed to produce mature tRNAs and instead accumulated the IVS–3′ exon intermediate (Fig. 4c and d; Fig. S1). Experimental confirmation of accurate intron excision at position 35/36 supported the reliability of our *in silico* protocol proposed above for identifying intron-containing tRNAs. Currently, the retention rate of tRNA genes is widely used as an indicator of genome completeness in bacterial genome quality assessments (45). We propose a two-step approach that integrates intron-aware tRNA prediction tools (e.g., tFind or ARAGORN) with splicing-boundary verification through BLAST searches against intron-less tRNA homologs when accurate evaluation of genome completeness is required.

## Distribution of group I and II introns within tRNA genes across bacteria

Given that the protocol had been experimentally verified, we next applied our *in silico* protocol to a set of 4,934 complete bacterial genomes spanning 63 phyla from the GTDB

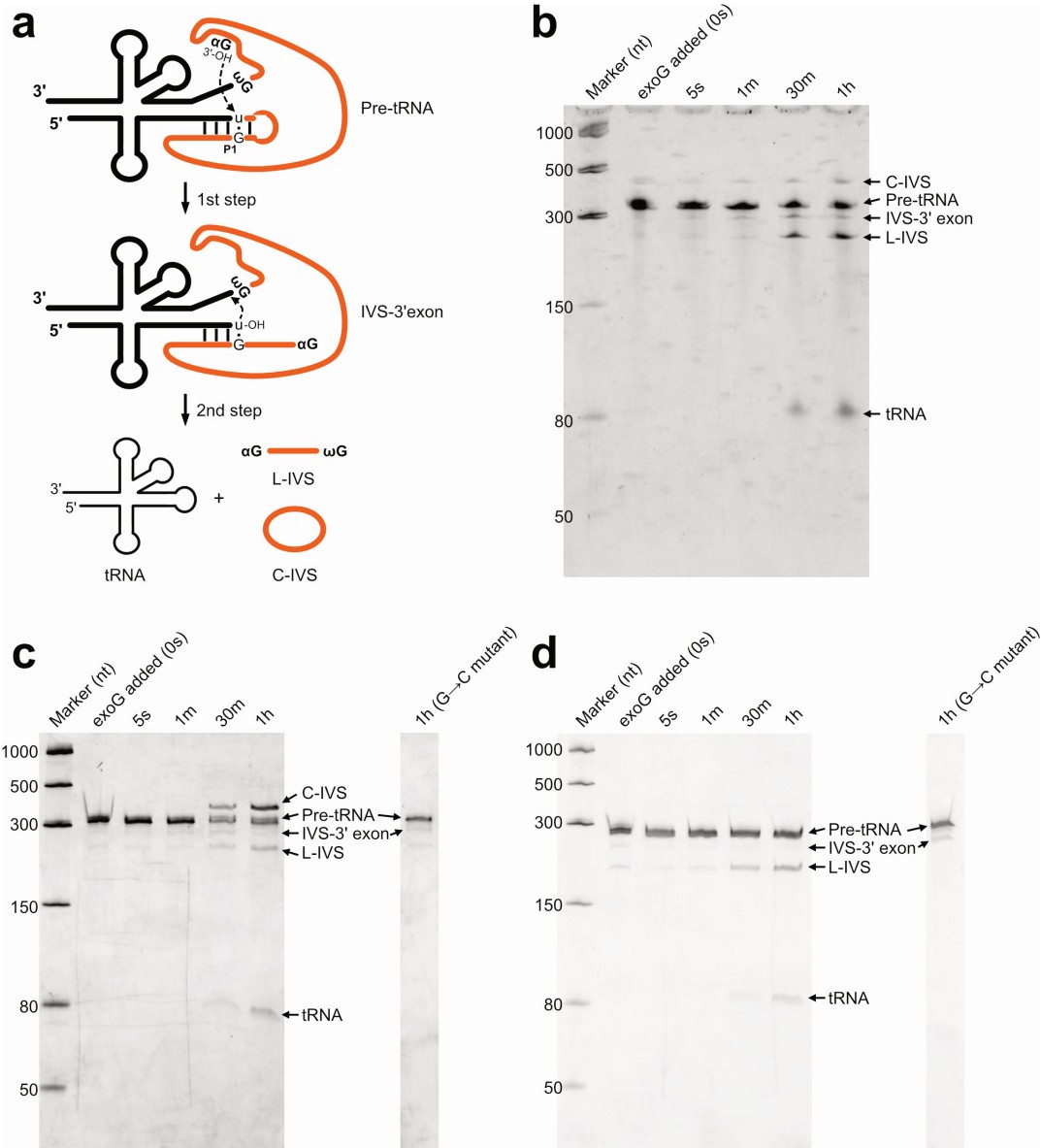

**FIG 4** Splicing activity of group I introns inserted at position 35/36 in tRNA genes of Patescibacteria. (a) Schematic drawing of the splicing reactions of group I introns in tRNA genes. ωG: a 3′-terminal guanosine of the intron. αG: an exogenous guanosine cofactor. L-IVS: a linear intron. C-IVS: a circular intron. (b) Gel electrophoresis of splicing reactions of the tRNA$^{Leu}$ containing a group I intron from the Cyanobacteriota genome (GenBank ID: BA000019.2). (c) Gel electrophoresis of splicing reactions of the tRNA$^{Asn}$ containing a group I intron from the Patescibacteria genome (accession: GCA_016700035.1). (d) Gel electrophoresis of splicing reactions of the tRNA$^{Asp}$ containing a group I intron from the Patescibacteria genome (accession: GCA_001029755.1). (b–d) Time points after GTP addition are indicated above each lane (0 seconds, 5 seconds, 1 minute, 30 minutes, and 1 hour). (c and d) Both original constructs based on the genome sequences and mutant constructs (3′ terminal G→C substitution in the group I intron) are shown; however, for the mutant constructs, only the results after 1 hour of splicing are presented. For each tRNA gene, the original and mutant constructs were assayed at the same concentration and loaded on the same gel.

r220 data set. This analysis identified a total of 333 tRNAs harboring group I introns across 18 bacterial phyla (Table S5). These included all tRNA$^{Asn}$ (GUU) and tRNA$^{Asp}$ (GUC) genes in Patescibacteria, which had not been accurately annotated when using tFind alone (Table 1). We then analyzed the insertion positions of these group I introns within tRNA genes. In agreement with previous studies (28–30, 41), insertions were commonly found at positions such as 33/34 in tRNA$^{fMet}$, 34/35 in tRNA$^{Leu}$, and 36/37 in tRNA$^{Arg}$ (Fig.

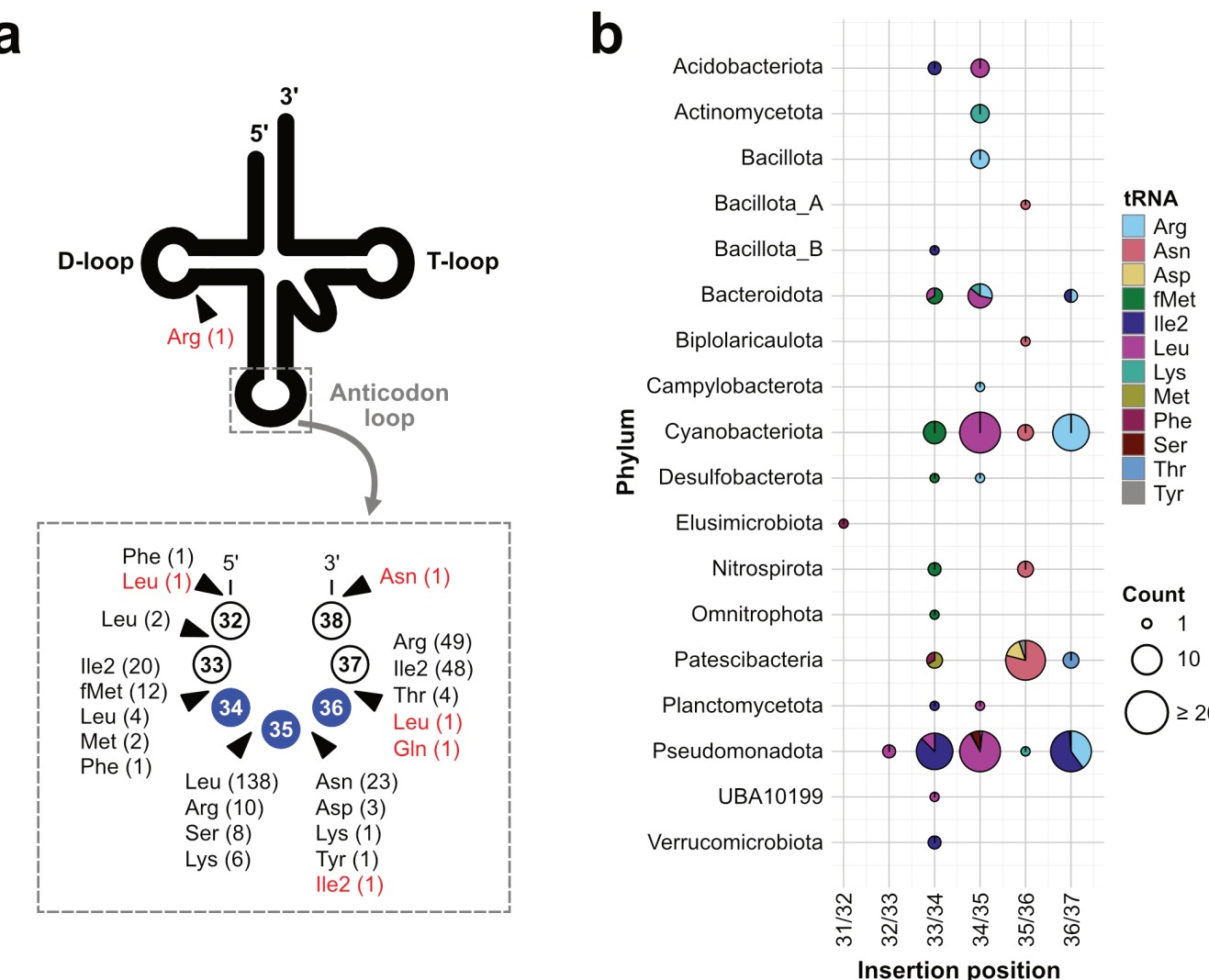

**FIG 5** Diversity of insertion sites of introns in bacterial tRNA genes. (a) Schematic representation showing the various insertion sites of group I and group II introns within the anticodon loop of tRNA genes. Group I introns are shown in black, and group II introns in red. Numbers in parentheses represent the number of introns identified at each position for each amino acid-specific tRNA gene. One group II intron was also found inserted in the D-loop. (b) Bubble chart showing the insertion positions of group I introns within tRNA genes across bacterial phyla. Each circle represents the number of genomes with introns inserted at a specific position (x-axis) and in a particular phylum (y-axis). Circle size corresponds to the number of observations (legend at bottom right). Colors indicate the amino acid of the associated tRNA genes.

5a). Beyond these well-characterized sites, we found that group I introns were inserted at nearly all positions within the anticodon loop except for position 37/38.

When we examined the distribution of insertion positions by phylum, introns in the phyla Pseudomonadota and Cyanobacteriota were mainly found at positions 33/34, 34/35, and 36/37, whereas those in the phyla Patescibacteria were strongly enriched at 35/36 (Fig. 5b). Of 28 tRNA introns inserted at the position 35/36, 19 were derived from Patescibacteria, while the remaining cases were found in the phyla Cyanobacteriota and Nitrospirota (3 each), and in Baccilota_A, Bipolaricaulota, and Pseudomonadota (1 each). These findings indicate that intron insertions at the position 35/36 were particularly enriched in Patescibacteria, distinguishing Patescibacteria from other bacterial phyla.

We further extended our analysis to examine the presence of group II introns within tRNA genes, although no instances of such insertions in bacterial tRNA genes have been reported to date. Among the tRNA intron sequences detected by ARAGORN, group II intron sequences were detected in only six loci across 4,934 complete bacterial genomes

(Fig. 5a; Table S5). Five of these insertions were in the anticodon loop and one in the D-loop. The insertion at positions 35/36 of the anticodon loop was found in a tRNA$^{Ile2}$ gene from a Cyanobacteriota genome, whereas the remaining five were detected in a single Bacillota_I genome (GCA_027945135.1). These results indicate that, although extremely rare, group II introns can also insert into tRNA genes, and their insertion sites are not limited to the anticodon loop but can extend to other regions such as the D-loop.

## Low prevalence of homing endonuclease-encoding tRNA introns in bacteria

Homing endonuclease genes (HEGs) are known to drive intron homing into intron-less alleles and thereby promote intron mobility. The presence of HEGs may thus facilitate the spread and persistence of these introns across bacterial lineages, potentially including horizontal transfer (36).

As group I introns longer than 500 bp are known to potentially encode HEGs (46), we analyzed the length distribution of the identified group I introns and found that their lengths ranged from 184 to 894 base pairs, with an average length of 257 bp (Fig. 6a). Only 12 of the 333 introns exceeded 500 bp, and all these encoded HEGs (Table S6). Group I introns within tRNA genes decoding the same anticodon tended to encode endonucleases from the same family. For example, three introns at position 36/37 in tRNA$^{Arg}$ (CCU) from Cyanobacteriota encoded LAGLIDADG-type endonucleases (PF03161); two introns at position 33/34 in tRNA$^{fMet}$ (CAU) from Cyanobacteriota encoded PD-(D/E)XK domain-containing endonucleases (PF11645); and four introns at position 35/36 in tRNA$^{Asn}$ (GUU) from Patescibacteria encoded endonucleases of the HNH family (PF01844) (Fig. 6a). Group II introns are known to encode intron-encoded proteins with reverse transcriptase activity (25), and such a protein was identified in one of the six tRNAs containing a group II intron (Table S6).

The HEGs found in rRNA introns of Patescibacteria belong to the LAGLIDADG family (33), suggesting that tRNA and rRNA introns may have been independently maintained through distinct evolutionary routes, and most tRNA-inserted group I introns did not encode HEGs, suggesting limited autonomous mobility. Although HEG-containing introns may be underdetected, this pattern indicates that horizontal transfer of tRNA introns is likely infrequent.

## Distinct structural subgroup of group I introns in Patescibacteria

We further analyzed the structural features of these introns. We classified the structural subgroups of each intron using Infernal with covariance models representing group I intron subgroups (46). Out of the 333 tRNA introns identified across bacteria, 81% belonged to the IC subgroup, and 25 of 28 introns inserted at the position 35/36 were classified into the IA (Fig. 6b). IA introns are rare but have been reported in certain bacteria, protists (such as centrohelids and cryptophytes), fungi, and viruses (46). While IC introns are generally the most abundant across all domains of life (46, 47) as well as in plastid tRNA$^{Leu}$ genes (47), none of the group I introns in tRNA genes from Patescibacteria were assigned to the IC, suggesting an evolutionary trajectory distinct from other bacterial phyla.

To investigate the evolutionary relationships among these introns, we constructed a maximum-likelihood phylogenetic tree using all bacterial tRNA intron sequences. The resulting phylogenetic tree revealed that the introns were broadly separated into the IA and IC subgroups (Fig. S2). Within the IA clade, tRNA$^{Asn}$ and tRNA$^{Asp}$ introns from Patescibacteria form a paraphyletic group relative to introns in tRNAs from other phyla, such as Pseudomonadota and Cyanobacteriota (Fig. 6c). Interestingly, based on the phylogeny of the IA subgroup, the three tRNA$^{Thr}$ introns of Patescibacteria (referred to as "divergent group I introns" [10]) exhibited longer branch lengths than the others but were supported by high bootstrap values together with tRNA introns of Patescibacteria (Fig. 6c), suggesting that tRNA$^{Thr}$ introns may have been derived from these introns.

The tRNA$^{Asn}$ introns from Patescibacteria, some of which encoded HEGs, were phylogenetically close to tRNA$^{Ile2}$ and tRNA$^{Leu}$ introns from Pseudomonadota (Fig. 6c).

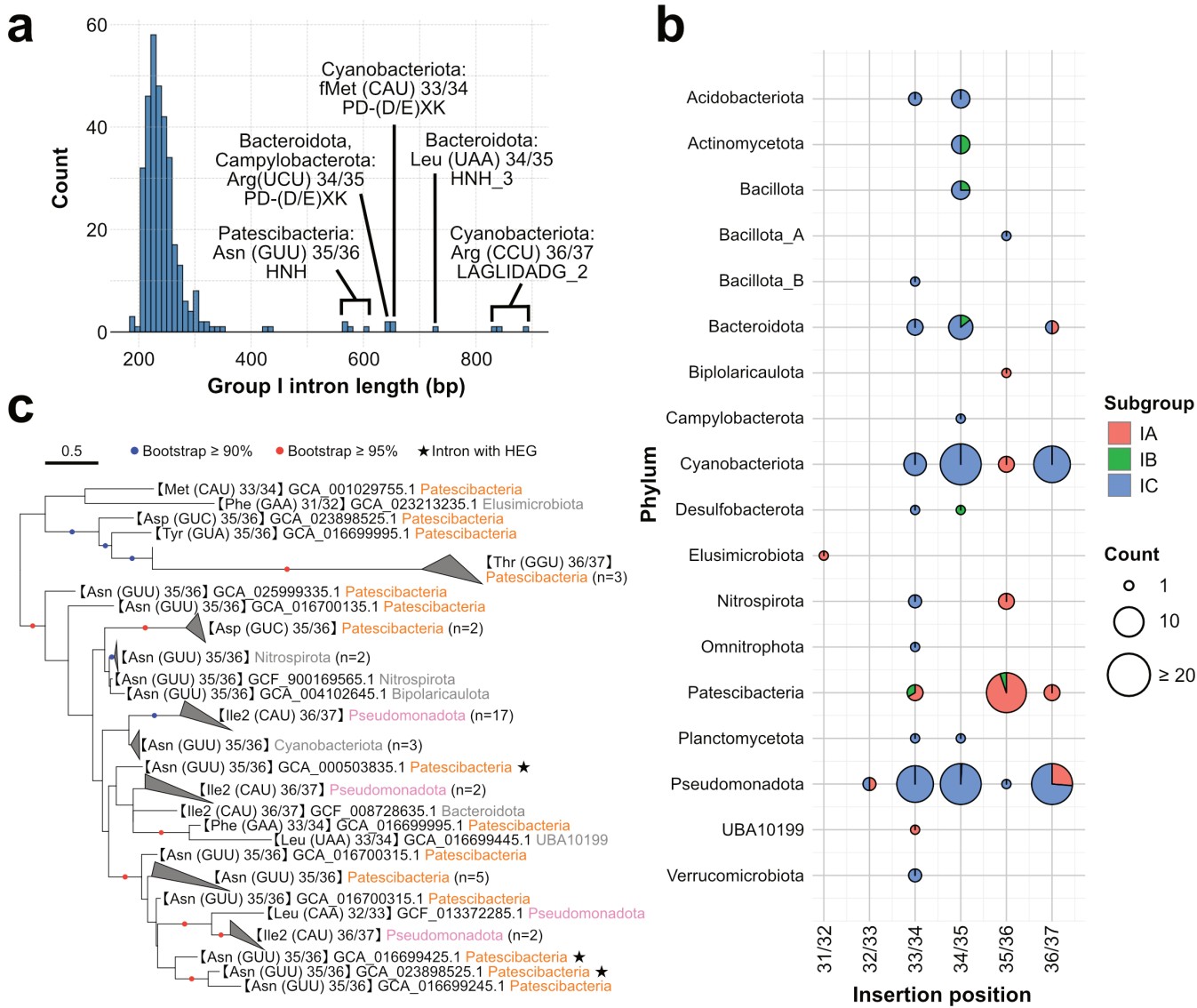

**FIG 6** Distribution of subgroups of group I introns across insertion positions and bacterial phyla. (a) Histogram showing the length distribution of group I introns identified from bacterial tRNA genes. Introns longer than 500 bp contained HEGs. Among these, HNH-family HEGs were identified in the tRNA$^{Asn}$ (GUU) of Patescibacteria and the tRNA$^{Leu}$ (UAA) of Bacteroidota; the PD-(D/E)XK family was found in the tRNA$^{fMet}$ (CAU) of Cyanobacteriota and the tRNA$^{Arg}$ (UCU) of Bacteridota and Campylobacterota; and the LAGLIDADG family was observed in the tRNA$^{Arg}$ (CCU) of Cyanobacteriota. (b) Bubble chart illustrating the distribution of subgroups of group I introns inserted at different positions within tRNA genes across bacterial phyla. Each circle represents the number of introns at a given insertion site (x-axis) in a particular phylum (y-axis), with the size of the circle indicating the count. The color within each circle denotes the proportion of subgroups: IA (red), IB (green), and IC (blue). (c) The phylogenetic tree was inferred using IQ-TREE with the TNe+G4 model selected by ModelFinder, based on 143 nucleotide sites from 55 sequences classified in the IA subgroup. Branches with bootstrap values between 90% and <95% are indicated by blue dots, and those with bootstrap values ≥95% are indicated by red dots. Tip labels include the corresponding tRNA genes and insertion sites, genome IDs, and phylum-level taxonomic annotations. Stars indicate group I introns that contain HEGs.

Based on our data, group I introns inserted at position 35/36 and classified in the IA subgroup that encode HEGs were found only in Patescibacteria. This raises the possibility that these introns were horizontally transferred from Patescibacteria to other phyla, or alternatively, that they were horizontally acquired from outside Patescibacteria and subsequently lost their homing endonucleases in other lineages. In this study, our analysis was limited to complete genomes to obtain a solid understanding of intron distribution; however, expanding the data set to include draft genomes may reveal

additional group I introns that encode HEGs, potentially allowing for more comprehensive phylogenetic analyses.

## Distribution of group I introns outside tRNAs in bacteria

Finally, to investigate how group I introns are distributed outside of tRNAs, we conducted a comprehensive search for group I introns. We analyzed group I introns across 4,934 complete genomes using Infernal with two covariance models, RF00028.cm and gpI_bact_tRNA.cm, applying a threshold of <1e-4. Because some group I introns were detected by only one of the two models (Fig. S3), both models were used to ensure comprehensive detection. As a result, a total of 846 group I introns were identified in 519 genomes across 30 phyla (Table S7). On average, genomes containing group I introns harbored 1.6 copies (standard deviation = 1.7, median = 1), indicating that most genomes conserved only a single intron. Group I introns were particularly enriched in the phyla Patescibacteria and Cyanobacteriota, with nearly 40% of the analyzed genomes retaining at least one group I intron (Fig. 7a).

As a comparison to group I introns, we also investigated the distribution of group II introns using RF00029.cm and identified a total of 5,662 group II introns in 1,091 genomes across 39 phyla. The average copy number per genome was 5.2 copies (standard deviation = 13.4, median = 2) (Table S8), suggesting a broader distribution and higher copy number per genome compared to group I introns. While the phylum Cyanobacteriota, which frequently contained group I introns, also showed a high prevalence of group II introns, no group II introns were detected in Patescibacteria (Fig. 7a). We cannot completely rule out the existence of a small number of highly divergent group II introns that may not be detectable in this analysis. However, because RF00029.cm targets the conserved catalytic core of group II introns and performs robustly across diverse bacteria (Fig. S3), our results suggest that canonical group II introns are genuinely uncommon in Patescibacteria.

To identify the functional gene categories that harbor group I introns in bacterial genomes, we classified the host genes of all identified introns. tRNA genes were the most frequent targets of intron insertion, accounting for 39% of all identified introns (Fig. 7b). Additional insertions occurred in tmRNA genes (6%), rRNA genes (13%), and protein-coding sequences (CDSs; 17%) (Table S7 and S9). Interestingly, 11% of introns were identified near TnpB-like genes (Table S7). IStrons are genetic elements that combine a group I intron with a mobile insertion sequence element and are known to carry TnpB genes encoding RNA-guided DNA endonucleases (48, 49). Therefore, the introns found adjacent to TnpB-like genes were presumed to be associated with IStrons.

Group I introns in tRNA genes were identified across 18 of the 63 bacterial phyla analyzed, representing a phylogenetic diversity comparable to that of rRNA introns, which were found in 15 phyla (Fig. 7c; Table S7). In Patescibacteria, group I introns are conserved across the entire phylum in both tRNA and rRNA genes, suggesting the presence of selection pressures or regulatory mechanisms that maintain these elements in this phylum. Although CDSs were the second most common insertion targets after tRNA genes, this was largely due to a few genomes harboring multiple intron-containing CDSs, most of which were found in the phylum Bacillota (Fig. 7c). Similarly, almost all tmRNA-associated introns were also observed in Bacillota (Fig. 7c). These results demonstrate that group I introns exhibit broad phylogenetic distribution in tRNA and rRNA genes, whereas their presence in CDS and tmRNA genes is more restricted.

## Perspective on the unusually high occurrence of tRNA introns in Patescibacteria

Group I introns are generally known to predominantly insert into structured RNAs such as rRNAs and tRNAs (36), and our observations in Patescibacteria are consistent with this tendency. Group II introns are generally associated with mobile genetic elements or intergenic regions and rarely integrate into essential genes (50). However, no group II introns were detected in the genomes of Patescibacteria. This difference in the retention

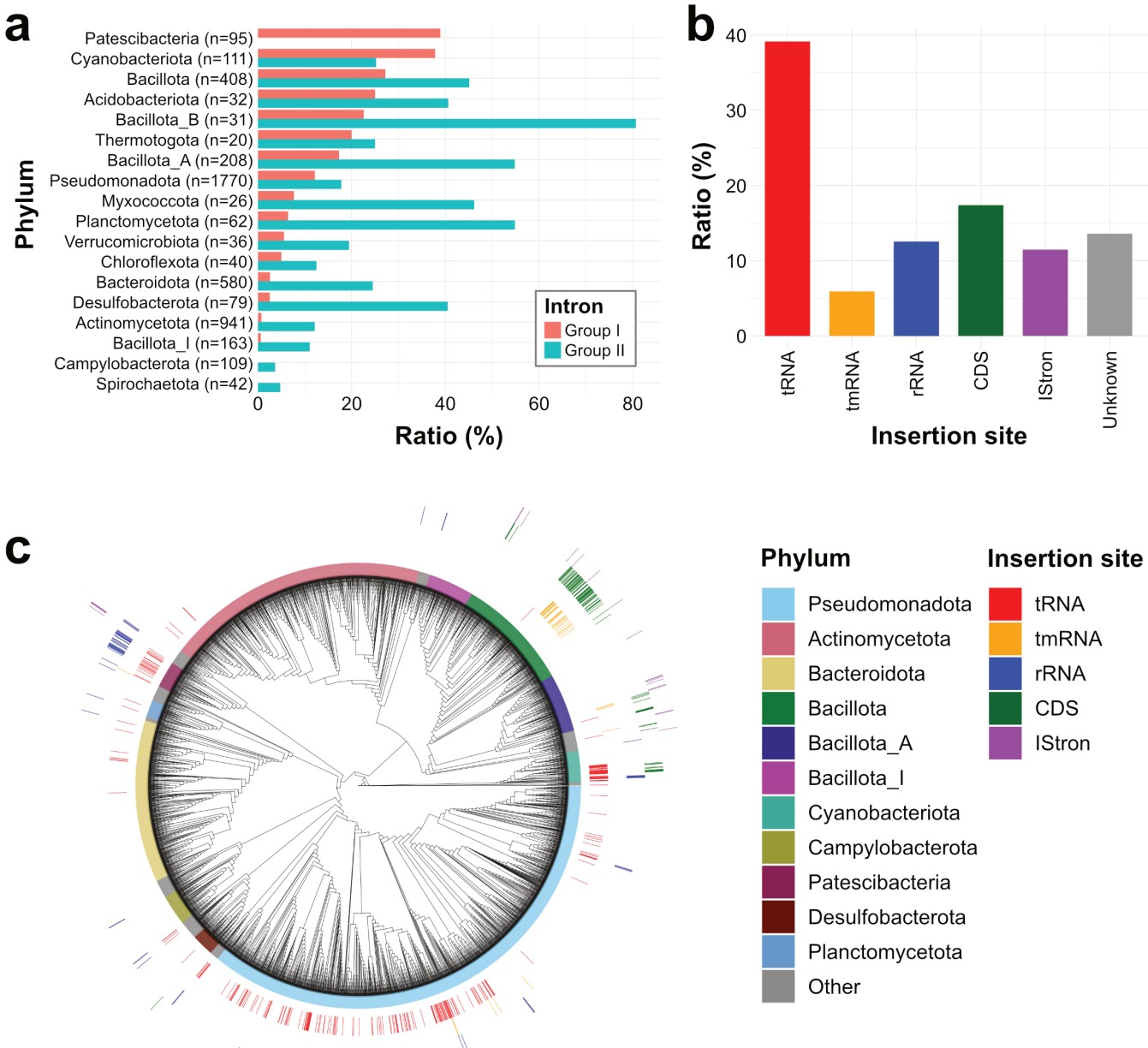

**FIG 7** Distribution of insertion sites of group I introns across bacterial genomes. (a) Bar plot showing the proportion of genomes containing group I/II introns for each phylum. Only phyla with ≥20 genomes are included. The retention rate was calculated as the number of genomes with group I/II introns divided by the total number of genomes in each phylum. (b) Bar graph showing the proportions of the 846 group I introns identified from bacterial genomes by their insertion sites: tRNA genes, tmRNA genes, rRNA genes, protein-coding sequences (CDSs), IStrons-like, and unknown loci. (c) Phylogenetic tree of 4,934 bacterial genomes constructed using concatenated alignments of bac120 marker genes. The tree was inferred using IQ-TREE with the Q.pfam+R10 model selected by ModelFinder, based on 5,035 amino acid positions. Phylum and insertion sites were mapped on the tree.

between group I and group II introns in Patescibacteria may reflect differences in splicing efficiency and cellular resource constraints. Group I introns are self-splicing ribozymes that catalyze their own excision through relatively simple RNA-based mechanisms. In contrast, group II intron splicing requires intron-encoded protein cofactors that assist in RNA folding, catalysis, and mobility (51). This protein-dependent process is mechanistically more complex and energetically demanding, and its efficiency is typically lower than that of group I introns (50). Given the highly reduced genomes and limited

metabolic capacities of Patescibacteria, such energetically costly systems may have been selectively lost or never established in these lineages.

From the point of genome replication efficiency, one might expect that Patescibacteria, like other bacteria, would have further streamlined their genomes by eliminating these introns. However, this has not occurred. The fact that these seemingly dispensable introns were retained despite extensive genome reduction is particularly intriguing. One possible explanation is structural. In both Patescibacteria and other bacteria, it was rare to find both intron-containing and intron-less copies of tRNA genes encoding the same anticodon within a single genome. Because even a single nucleotide change in the anticodon can alter codon recognition, introns within these tRNA genes may be difficult to lose without compromising translational fidelity, which would likely be lethal.

Another possibility is physiological. The preferential retention of introns in specific bacterial lineages such as Patescibacteria and Cyanobacteriota may reflect physiological or regulatory advantages. Stable introns might influence RNA folding or the maturation process of essential RNAs under particular environmental or metabolic conditions. Patescibacteria have extensively streamlined their genomes, losing many transcriptional regulators. The persistence of group I introns may thus supply a compensatory RNA-level control that fine-tunes expression and translation.

Future studies integrating comparative transcriptomics, RNA structural analysis, and functional assays will be essential to determine whether group I introns actively participate in regulatory networks and to what extent RNA-based regulation substitutes for protein-mediated control in Patescibacteria and related lineages.

## MATERIALS AND METHODS

### Data set

From all representative bacterial genomes defined in GTDB r220 (38), we downloaded 4,934 genomes that met the following criteria: (i) assembly level: complete genome; (ii) number of contigs: 1; and (iii) contamination: <5%.

### Identification of group I/II introns and rRNA genes

Group I introns, group II introns, and rRNA genes were detected using cmsearch or cmscan from Infernal v.1.1.4 (52) against covariance models, with a threshold e-value of <1E-4. We used RF00028.cm and gpI_bact_tRNA.cm (41) for group I introns, RF00029.cm for group II introns, RF00177.cm for bacterial SSU rRNA, and RF02541.cm for bacterial LSU rRNA. All covariance models were obtained from the Rfam database (53), except for gpI_bact_tRNA.cm. For downstream analyses of group I introns, when the same genomic locus was detected by both covariance models, only a single hit was retained by selecting the hit with the higher score to avoid redundancy.

### Identification of tRNAs and intron insertions

To evaluate the detection of the complete set of tRNA genes in Patescibacteria genomes, we used tRNAscan-SE 2.0 with the -B option (39), ARAGORN v.1.2.41 with the -i option (40), and tFind with the default setting (41).

To identify tRNAs containing group I introns in Patescibacteria genomes, we performed the following steps. First, intron-less tRNA sequences detected by tRNAscan-SE 2.0 were used to build a BLAST database with the makeblastdb command from BLAST+ v.2.12 (54). Predicted intron regions from Infernal (RF00028.cm) were used as the core query sequences, with an additional 400 bp extracted upstream of the predicted intron start and downstream of the predicted intron end. BLAST searches were performed using the blastn command with the -task blastn-short option. When tRNA sequences were detected on both sides of the intron, we identified the anticodon loop based on tRNA secondary structure and defined the group I intron boundaries as a region that (i) formed a 7-nt anticodon, (ii) started immediately after a U, and (iii) ended

at a G. After determining the boundaries, we reanalyzed the spliced sequence using tRNAscan-SE to verify whether a valid tRNA secondary structure could be formed. We also performed another BLAST search (-task blastn-short) using the full-length spliced tRNA sequence and confirmed that its anticodon matched that of the top-scoring intron-less homolog (alignment coverage >75%). If the group I intron had an ambiguous splicing boundary (i.e., multiple boundary combinations were possible), we referred to the top-scoring intron-less homolog in the second BLAST step to select the most appropriate one.

For the identification of tRNAs containing group I introns across all bacterial genomes, we applied a two-step approach that integrates tFind with splicing-boundary verification through BLAST searches against intron-less tRNA homologs. Intron-less tRNA sequences detected from all bacterial genomes by tRNAscan-SE 2.0 were used to build a BLAST database with the makeblastdb command. We performed a BLAST search (-task blastn-short; alignment coverage >75%) with the tRNAs predicted by tFind and confirmed that their anticodons matched those of the top-scoring intron-less homologs. When the anticodons of tRNAs predicted by tFind did not correspond to those of the intron-less homologs, we adjusted the splicing boundary as described above. For tRNAs with the CAU anticodon (i.e., Met, fMet, and Ile2), we determined the amino acid assignment according to the top-scoring intron-less homologs.

For the identification of tRNAs containing group II introns, we first detected tRNA intron sequences using ARAGORN and then examined whether these regions overlapped with group II intron sequences detected by Infernal across bacterial genomes. For tRNA candidates containing group II introns, BLAST searches were performed as described above, and the splicing boundaries were adjusted based on the BLAST alignment results.

## Identification of rRNA, tmRNA, and protein-coding genes containing group I introns

The intron regions, along with 1,000 bp upstream and downstream flanking sequences, were used as queries for a cmsearch using Rfam covariance models for bacterial SSU rRNA (RF00177.cm), LSU rRNA (RF02541.cm), and tmRNA (RF00023.cm). The --anytrunc option was used to improve the detection of fragmented sequences. If rRNA sequences were detected on both sides of a group I intron, the intron was considered to be inserted within an rRNA gene. In the case of tmRNA, group I intron insertion can cause the 3′ end of the tmRNA to become too short to be detected by Infernal (55). Therefore, when only a partial tmRNA sequence was detected near one side of the intron, we checked the intron boundary at the T loop and verified the presence of a matching tmRNA sequence on the opposite side before concluding that an insertion had occurred. We also used tFind to predict intron-containing tmRNAs and merged the result with our own prediction (Table S9).

To determine whether introns were inserted within protein-coding genes, intron regions together with 4,000 bp flanking sequences on both upstream and downstream sides were analyzed using the blastx command in DIAMOND (56). When homologous sequences derived from the same gene were detected on both sides of an intron, the intron was classified as being inserted within a CDS. The blastx searches were performed in two steps. In the first step, the NCBI non-redundant protein database was used to identify genes frequently found adjacent to introns (Table S10). Consistent with previous reports, this set included genes previously reported to be disrupted by intron insertions, such as flagellin and ribonucleotide reductase (57–60). In the second step, a custom database was constructed by retrieving these genes from Swiss-Prot entries curated in InterPro (61) to identify introns that interrupt CDSs. When TnpB-related genes were detected in the vicinity of an intron, the intron was classified as an IStron-like intron (48, 49).

## Secondary structure prediction and visualization of group I introns in tRNA$^{Asn}$ and tRNA$^{Asp}$ from Patescibacteria

The secondary structure diagrams of group I introns in tRNA$^{Asn}$ and tRNA$^{Asp}$ from Patescibacteria (GCA_016700035.1 and GCA_001029755.1, respectively) shown in Fig. 2 were generated based on visualizations produced by R2DT (62). For a small number of stems that were not predicted by R2DT (e.g., the P7.1 helix in Fig. 2a), we manually refined the diagrams based on established group I intron secondary structure models reported in previous studies (36, 63, 64).

### In vitro transcription and splicing assay

DNA templates for *in vitro* transcription were synthesized by Twist Bioscience and Eurofins (listed in Table S4) and PCR-amplified using PrimeSTAR MAX DNA Polymerase (TaKaRa). For PCR amplification, a sequence containing the T7 promoter (5′-TAATACG ACTCACTATA-3′) was added to the 5′ end of the forward primer, allowing the target sequence to be positioned immediately downstream of the promoter (see Table S4 for details). For the tRNA$^{Asn}$ construct from GCA_016700035.1, an additional guanine (G) was inserted between the T7 promoter and the tRNA sequence to enhance transcription initiation efficiency. For the negative control, a reverse primer was designed to substitute the predicted 3′ terminal guanine of the group I intron with cytosine. PCR products were purified using the NucleoSpin Gel and PCR Clean-up kit (TaKaRa). All purified PCR products were subjected to Sanger sequencing to confirm the sequence accuracy.

*In vitro* transcription was performed using T7 RNA polymerase (New England Biolabs) following the manufacturer's protocol. Following incubation at 37°C for 2 hours, 0.2 µL DNase I (Nippon Gene) was added and incubated for an additional 15 minutes at 37°C. Transcribed RNAs were purified using the RNA Clean & Concentrator kit (Zymo Research).

Precursor RNAs were resolved on a 7% UREA-PAGE gel (65) by electrophoresis at 180 V. The gel was stained with SYBR Gold (Thermo Fisher Scientific) and imaged using a LuminoGraph I (ATTO) equipped with a WSE-5600 CyanoView (505 nm). A precursor band corresponding to a tRNA with intron was excised from the gel and incubated in 0.3 M NaCl with rotation for over 12 hours to elute the RNA (66). The eluted product was filtered and purified by ethanol precipitation.

Splicing assays were performed with reference to previous studies (24, 29). Prior to splicing, RNA samples were denatured at 95°C for 3 minutes. Samples were then incubated at 30°C in buffer (50 mM HEPES-KOH, pH 7.5, 5 mM MgCl$_2$, RNase-free water to 10 µL) containing 20 nM–200 nM RNA. Splicing was initiated by the addition of 0.1 mM GTP and terminated at 5 seconds, 1 minute, 30 minutes, and 1 hour by adding 1 µL of 0.3 M EDTA. The concentrations of the original and negative control RNAs were adjusted to the same level, and electrophoresis was performed on a 7% UREA-PAGE gel as described above, followed by staining and imaging.

### Validation of splicing by RT-PCR and sequencing

RNA products corresponding to the predicted mature tRNA band were excised and purified from the gel after a 1 hour incubation in the splicing assay, following the same procedure described above. Reverse transcription was performed using PrimeScript Reverse Transcriptase (TaKaRa) according to the manufacturer's protocol, with random primers. Prior to the reaction, the RNA templates were denatured at 95°C for 3 minutes.

To enable sequencing of the tRNA sequence (less than 80 bp) by Sanger sequencing, PCR amplification of the cDNA was performed (see Table S4) with PrimeSTAR MAX DNA Polymerase. The PCR products were then further amplified using the forward primer (5′-GCTATTTAGGTGACACTATAG-3′) and the reverse primer (5′-AATACGACTCACTATAGG -3′), and the resulting fragments were gel-purified using the NucleoSpin Gel and PCR Clean-up kit. The purified products were confirmed by Sanger sequencing.

## Construction of phylogenetic tree for bacterial genomes

Phylogenetic analysis was conducted using bac120 marker gene alignments provided by GTDB r220. Maximum-likelihood phylogenetic trees were constructed using IQ-TREE v.2.1.4-beta with a model selected by ModelFinder and 1,000 ultrafast bootstrap replicates (67). The phylogenetic tree was visualized using ggtree (68).

## Detection of genes encoding homing endonucleases and reverse transcriptase

To identify potential protein-coding genes within group I introns longer than 500 bp and within group II introns, open reading frames (ORFs) were predicted from intron sequences using ORFfinder (https://www.ncbi.nlm.nih.gov/orffinder/). The predicted ORFs were analyzed with hmmscan from HMMER v.3.3.2 (http://hmmer.org/) against the Pfam-A.hmm v.37.0 (69) (E-value <0.01) to detect conserved homing endonuclease or reverse transcriptase domains.

## Subgroup classification of group I introns

Subgroups classification was performed following the approach described in a previous study (46), using Infernal (cmscan with the options --max and --tblout). Covariance models for group I intron subgroups were downloaded from https://github.com/LaraSellesVidal/Group1IntronDatabase. The covariance model that yielded the highest alignment score was used to assign the subgroup for each intron.

## Construction of phylogenetic tree for group I introns

Group I intron sequences inserted into tRNA genes were aligned using cmalign from Infernal with the gpI_bact_tRNA.cm. To retain only well-aligned and conserved regions, the resulting alignment was trimmed using trimAl with the -gt 0.9 option (70). Maximum-likelihood phylogenetic trees were then constructed using IQ-TREE v.2.1.4-beta with a model selected by ModelFinder and 1,000 ultrafast bootstrap replicates. The phylogenetic tree was visualized using iTOL (71).

### ACKNOWLEDGMENTS

This study was supported by JST CREST JPMJCR20S4, SPRING JPMJSP2108, and JSPS KAKENHI 24K00749, and S.S. was partly supported by JSPS KAKENHI 22H05152 and the AMED ASPIRE Program 25jf0126009h0002. Computation time was provided by HOKUSAI, RIKEN.

### AUTHOR AFFILIATIONS

[1]Department of Integrated Biosciences, Graduate School of Frontier Sciences, The University of Tokyo, Kashiwa, Chiba, Japan
[2]Geobiology and Astrobiology Laboratory, RIKEN Pioneering Research Institute, Wako, Saitama, Japan
[3]Institute of Space and Astronautical Science (ISAS), Japan Aerospace Exploration Agency (JAXA), Sagamihara, Kanagawa, Japan

### AUTHOR ORCIDs

Yuna Nakagawa  http://orcid.org/0009-0000-3112-0660
Kazuaki Amikura  http://orcid.org/0000-0001-9813-9462
Kimiho Omae  http://orcid.org/0009-0001-4033-5177
Shino Suzuki  http://orcid.org/0000-0003-2176-2103

## FUNDING

| Funder | Grant(s) | Author(s) |
|---|---|---|
| Japan Science and Technology Agency | JPMJCR20S4 | Shino Suzuki |
| Japan Science and Technology Agency | JPMJSP2108 | Yuna Nakagawa |
| Japan Society for the Promotion of Science | 24K00749 | Kazuaki Amikura |
| Japan Society for the Promotion of Science | 22H05152 | Shino Suzuki |
| Japan Agency for Medical Research and Development | 25jf0126009h0002 | Shino Suzuki |

## AUTHOR CONTRIBUTIONS

Yuna Nakagawa, Conceptualization, Data curation, Formal analysis, Funding acquisition, Investigation, Methodology, Validation, Visualization, Writing – original draft, Writing – review and editing | Kazuaki Amikura, Conceptualization, Funding acquisition, Investigation, Methodology, Validation, Writing – review and editing | Kimiho Omae, Formal analysis, Methodology, Visualization, Writing – review and editing | Shino Suzuki, Conceptualization, Funding acquisition, Methodology, Project administration, Supervision, Writing – original draft, Writing – review and editing

## ADDITIONAL FILES

The following material is available online.

### Supplemental Material

**Supplemental material (mSystems01536-25-S0001.pdf).** Figures S1-S3 and legends for Tables S1-S10.
**Supplemental tables (mSystems01536-25-S0002.xlsx).** Tables S1-S10.

### Open Peer Review

**PEER REVIEW HISTORY (review-history.pdf).** An accounting of the reviewer comments and feedback.

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
