## [Reviewer comments · mSystems]

Group I Introns in tRNA Genes of Patescibacteria

Yuna Nakagawa, Kazuaki Amikura, Kimiho Omae, and Shino Suzuki

Corresponding Author(s): Shino Suzuki, Rikagaku Kenkyujo

Review Timeline:

Submission Date:	October 29, 2025
Editorial Decision:	December 19, 2025
Revision Received:	January 1, 2026
Accepted:	January 7, 2026

Editor: Nicola Vitulo

Reviewer(s): Disclosure of reviewer identity is with reference to reviewer comments included in decision letter(s). The following individuals involved in review of your submission have agreed to reveal their identity: Jakub Barylski (Reviewer #3)

Transaction Report:

DOI: <https://doi.org/10.1128/msystems.01536-25>

Re: mSystems01536-25 (Group I and II introns in tRNA Genes of Patescibacteria)

Dear Dr. Shino Suzuki:

While the reviews were generally positive, they still has a few issues that were not fully addressed in your response. Please strive to address these in your resubmission.

Revision Guidelines

Sincerely,
Nicola Vitulo
Editor
mSystems

Reviewer #2 (Comments for the Author):

In the revision of this manuscript, the authors presented their findings of group I and II introns in tRNA genes of Patescibacteria. They included the use of tFind and BLAST to refine the identification method of group I intron, which is a large improvement from the previous submission of the study. The use of additional covariance model developed previously by Williams (2020) and the experimental validation also strengthen the findings. However, the authors still only based on R2DT and manual refinement for

predicting the secondary structure of group I introns. By default, R2DT only has RF00028 Rfam covariance model for group I intron. Unless the authors added in new models and templates, the secondary structure created by R2DT will likely to be incorrect. The authors need to provide more details on the manual refinement they made. For example, for the ones that were detected using Williams' model, did the authors use the secondary structure generated by Infernal to create the visualization? For the ones missed by both models, how did the authors determine the secondary structure?

In addition, the authors increased the emphasis on the detection of group II introns in tRNA genes by including them in the article title and a results section title. Yet, the authors could not find any group II intron in tRNA genes of Patescibacteria. This causes misleading information delivered by the article title "Group I and II Introns in tRNA Genes of Patescibacteria" as the readers will generally expect findings and/or characterizations of group II introns in Patescibacteria with such a title instead of a complete absence. I would suggest the authors reword the title to avoid this problem. Moreover, the method being used for group II intron detection has a similar problem as the detection of group I intron in the previous submission. Rfam covariance model RF00029 which was used to identify group II introns was trained with 92 sequences. Only 16 sequences were originated from bacteria, with 6 in Bacillota, 2 in Cyanobacteriota, and the rest in Proteobacteria. Therefore, it is not hard to understand that the small number of detected group II introns are in Bacillota and Cyanobacteriota genomes. If the group II introns in Patescibacteria are diverse from the training data, it is possible that the model may not be able to detect them. The authors should mention this limitation in the findings.

Reviewer #3 (Comments for the Author):

Generally the paper is skilfully written. I really enjoyed the introduction, unfortunately narration gets a bit convoluted in later sections. I am aware that authors need to explain complex experiments, but parts of the paper where they do that, could use some editing to increase the clarity and brevity. The findings may not be groundbreaking, but they are still interesting and important in the field. In fact, my intuition is, that authors touched a much bigger subject and if they pursue the topic and broaden the scope, the paper could have significantly larger impact.

There are, however, some issues that need to be resolved before the paper is ready to be published:

109 - title "Undetected tRNA^{Asn} and tRNA^{Asp} Genes in Patescibacteria by Standard Annotation Tools" sounds weird. Maybe "tRNA^{Asn} and tRNA^{Asp} genes in Patescibacteria undetected by Standard Annotation Tools"?

124 - "Given that such unexpected absences were observed (Fig. 1a), we further examined to detect corresponding tRNA genes"; the syntax of this sentence is faulty. Additionally, can one really observe the absence?

233/234/236 - "... LAGLIDADG-type (...) PD-(D/E)XK (...) HNH ..."; add identifiers (InterPro/PFAM or other accession numbers) to unambiguously point to specific domains/families.

290 - "we also investigated the distribution of group II introns using the same method"; it may be important to remind the reader, that the method is the same, but applied CM is different (RF00029).

358 - "Nucleotide sequences of 4,934 representative bacterial genomes were downloaded from GTDB r220"; this number seems very low. Site <https://gtdb.ecogenomic.org/stats/r220> or [gtdb_genomes_reps_r220.tar.gz](https://gtdb.ecogenomic.org/genomes_reps_r220.tar.gz) include much more. What was the selection criterion?

364 - you need to explain what the model "gpi_bac_trna.cm" exactly is and provide a rationale of using it. It is far from clear in the current version of methods.

366 - "For group I introns, cmscan were also used, and when both models yielded hits for the same locus, the one with the higher score was retained"; Why? What is the rationale behind such approach?

377 - "Predicted intron regions from Infernal, including 400 bp upstream and downstream flanks, were used as queries"; Some introns can be larger. I'd consider extending the search scope further (e.g. to 1000 bp) to avoid selection bias (selection against longer nuclease-caring introns)

417-426 - "The intron regions and their 4,000 bp flanking sequences (...) as an IStron-like intron"; this section seems particularly convoluted and may need rewriting

475 - "Detection of Homing Endonucleases and Reverse Transcriptases"; I'd clarify why ORFs were used instead of genes (it makes sense but is not explained in the paper).

487 - "Construction of Phylogenetic Tree for Group I introns"; Are you sure MAFFT method is appropriate for divergent non-coding RNAs that often display structural rather than sequence similarity?

1 **Response to Reviewers**

**Reviewer 2**

*In the revision of this manuscript, the authors presented their findings of group I and II*
*introns in tRNA genes of Patescibacteria. They included the use of tFind and BLAST to*
*refine the identification method of group I intron, which is a large improvement from the*
*previous submission of the study. The use of additional covariance model developed*
*previously by Williams (2020) and the experimental validation also strengthen the findings.*

We sincerely appreciate the reviewer's valuable comments on our manuscript. We learned a lot
from this review process. Please find our point-by-point responses below.

*However, the authors still only based on R2DT and manual refinement for predicting the*
*secondary structure of group I introns. By default, R2DT only has RF00028 Rfam covariance*
*model for group I intron. Unless the authors added in new models and templates, the*
*secondary structure created by R2DT will likely to be incorrect. The authors need to provide*
*more details on the manual refinement they made. For example, for the ones that were*
*detected using Williams' model, did the authors use the secondary structure generated by*
*Infernal to create the visualization? For the ones missed by both models, how did the authors*
*determine the secondary structure?*

Thank you for pointing this out. We acknowledge that our description of R2DT in the original
manuscript may have caused confusion. In this study, R2DT was used only to generate
secondary-structure diagrams for visual inspection, not for de novo structure prediction, and it
was applied only to the two group I introns shown in Fig. 2 (GCA_016700035.1 and
GCA_001029755.1). For both introns, R2DT reproduced most of the major stem structures
typically observed in group I introns. For a small number of stems that were not predicted by
R2DT (e.g., the P7.1 helix in Fig. 2a), we manually refined the diagrams based on established
group I intron secondary-structure models reported in previous studies (Hauser et al., 2014,
Mobile DNA; Ikawa et al., 2000, Nat. Struct. Biol; Golden et al., 2005, Nat. Struct. Mol. Biol).
We did not perform secondary-structure prediction/visualization for introns detected only by the
Williams' model, nor for introns that were not detected by any covariance model. Both introns
shown in Fig. 2 were further experimentally validated by self-splicing assays, and the mature
tRNA sequences were verified by Sanger sequencing. We have revised the manuscript to clarify
the scope of the R2DT analysis (Page 20, Lines 425–431).

*In addition, the authors increased the emphasis on the detection of group II introns in tRNA*
*genes by including them in the article title and a results section title. Yet, the authors could*
*not find any group II intron in tRNA genes of Patescibacteria. This causes misleading*
*information delivered by the article title "Group I and II Introns in tRNA Genes of*
*Patescibacteria" as the readers will generally expect findings and/or characterizations of*
*group II introns in Patescibacteria with such a title instead of a complete absence. I would*
*suggest the authors reword the title to avoid this problem.*

We agree that original title could be misleading, as no group II introns were detected in tRNA
genes of Patescibacteria. To avoid this potential misunderstanding, we have revised the title to
“Group I Introns in tRNA Genes of Patescibacteria”.

*Moreover, the method being used for group II intron detection has a similar problem as the*
*detection of group I intron in the previous submission. Rfam covariance model RF00029*
*which was used to identify group II introns was trained with 92 sequences. Only 16 sequences*
*were originated from bacteria, with 6 in Bacillota, 2 in Cyanobacteriota, and the rest in*
*Proteobacteria. Therefore, it is not hard to understand that the small number of detected*
*group II introns are in Bacillota and Cyanobacteriota genomes. If the group II introns in*
*Patescibacteria are diverse from the training data, it is possible that the model may not be*
*able to detect them. The authors should mention this limitation in the findings.*

Thank you for this insightful comment. We acknowledge that covariance-model searches,
including RF00029.cm, may have reduced sensitivity for extremely divergent group II introns.
However, RF00029 targets the conserved catalytic core and performs robustly across diverse
bacteria: in our dataset, it detected 5,662 group II introns in 1,091 of 4,934 genomes spanning 39
phyla. Therefore, while we cannot completely exclude rare highly divergent elements below the
detection threshold, our results support the conclusion that canonical group II introns are
genuinely uncommon in Patescibacteria in the currently available genomes. We revised the
manuscript to clarify this limitation and the scope of our inference (Page 14, Lines 292–296).

**Reviewer 3**

*Generally the paper is skilfully written. I really enjoyed the introduction, unfortunately*
*narration gets a bit convoluted in later sections. I am aware that authors need to explain*
*complex experiments, but parts of the paper where they do that, could use some editing to*
*increase the clarity and brevity. The findings may not be groundbreaking, but they are still*
*interesting and important in the field. In fact, my intuition is, that authors touched a much*

***bigger subject and if they pursue the topic and broaden the scope, the paper could have***
***significantly larger impact. There are, however, some issues that need to be resolved before the***
***paper is ready to be published:***

We appreciate Reviewer 3's careful review and valuable comments. Our responses to each point
are provided below.

***109 - title "Undetected tRNAAsn and tRNAAsp Genes in Patescibacteria by Standard***
***Annotation Tools" sounds weird. Maybe "tRNAAsn and tRNAAsp genes in Patescibacteria***
***undetected by Standard Annotation Tools"?***

Thank you for pointing this out. As the original wording was unclear, we have revised the title
in accordance with your suggestion (Page 7, Line 106).

***124 - "Given that such unexpected absences were observed (Fig. 1a), we further examined to***
***detect corresponding tRNA genes"; the syntax of this sentence is faulty. Additionally, can one***
***really observe the absence?***

We agree that the original sentence was syntactically incorrect and that the expression "observe
the absence" was inappropriate. We have revised the manuscript to improve clarity and
readability (Page 7, Lines 124-125).

***233/234/236 - "... LAGLIDADG-type (...) PD-(D/E)XK (...) HNH ..."; add identifiers***
***(InterPro/PFAM or other accession numbers) to unambiguously point to specific***
***domains/families.***

Thank you for this suggestion. As requested, we have added the corresponding Pfam accession
numbers to unambiguously identify the domains (Page 12, Lines 232-236). In addition, detailed
information on the detected homing endonuclease genes is provided in Supplementary Table 6.

***290 - "we also investigated the distribution of group II introns using the same method"; it***
***may be important to remind the reader, that the method is the same, but applied CM is***
***different (RF00029).***

As suggested, we have revised the manuscript to clarify that a different covariance model
(RF00029.cm) was applied for the detection of group II introns (Page 14, Lines 286-288).

**358 - "Nucleotide sequences of 4,934 representative bacterial genomes were downloaded**
**from GTDB r220"; this number seems very low.**

**Site <https://gtdb.ecogenomic.org/stats/r220> or gtdb_genomes_reps_r220.tar.gz include much**
**more. What was the selection criterion?**

Thank you for pointing out this inaccurate description in the original manuscript. The number
4,934 does not refer to the total number of representative bacterial genomes available in GTDB
r220. Rather, we downloaded 4,934 genomes from the GTDB r220 representative bacterial
genome set after applying the following filters: (i) assembly level: complete genome; (ii) number
of contigs: 1; and (iii) contamination <5%. To assess the presence or absence of introns as
accurately as possible, our analysis was restricted to complete genomes. We have revised the
manuscript to clarify this point (Page 17, Lines 354–356).

**364 - you need to explain what the model "gpi_bac_trna.cm" exactly is and provide a**
**rationale of using it. It is far from clear in the current version of methods.**

Thank you for this comment. We now explicitly describe what gpI_bact_tRNA.cm is and why
we used it. gpi_bac_tRNA.cm is a bacterial tRNA/tmRNA-focused covariance model
implemented in tFind (Williams et al., 2020, bioRxiv), trained on 26 representative group I
introns inserted into bacterial tRNA and tmRNA genes. In contrast, the Rfam model RF00028
was built from 12 group I introns largely derived from phage, chloroplast, and eukaryotic
sequences, and is therefore not optimized for bacterial tRNA-inserted introns. Accordingly, we
used gpI_bact_tRNA.cm to improve detection of group I introns specifically inserted in
bacterial tRNA genes. We added this description and rationale to the Results section (Page 8,
Lines 152–154; Page 14, Lines 276–279) and slightly revised Fig. 1.

**366 - "For group I introns, cmscan were also used, and when both models yielded hits for the**
**same locus, the one with the higher score was retained"; Why? What is the rationale behind**
**such approach?**

In this study, group I introns were detected using two covariance models, RF00028.cm and
gpI_bact_tRNA.cm. In subsequent analyses, we examined the genomic insertion sites of group I
introns, including insertions in such as rRNA genes, and coding sequences. In many cases, the
same intron locus was detected by both covariance models. Although these hits corresponded to
the same genomic region, the predicted start and end positions occasionally differed by several

nucleotides between models.
For downstream analyses, which require a single set of coordinates for each intron, it was
necessary to retain only one hit per locus. Therefore, when both models detected the same locus,
we retained the hit with the higher Infernal score as a practical and consistent criterion to
remove redundancy, rather than as an indication that one model was biologically more
appropriate than the other. We have revised the manuscript to clarify this rationale and to avoid
potential misunderstanding (Page 17, Lines 362-365).

***377 - "Predicted intron regions from Infernal, including 400 bp upstream and downstream***
***flanks, were used as queries"; Some introns can be larger. I'd consider extending the search***
***scope further (e.g. to 1000 bp) to avoid selection bias (selection against longer nuclease-***
***caring introns)***

We acknowledge that the original wording was unclear and may have caused misunderstanding.
The 400 bp flanking regions do not include the intron sequence itself. Specifically, we examined
the regions extending from 400 bp upstream of the intron start predicted by Infernal to 400 bp
downstream of the predicted intron end. Therefore, long intron sequences themselves were fully
included in the analysis. To avoid further misunderstanding, we have revised the relevant text
accordingly (Page 18, Lines 372-375).

***417-426 - "The intron regions and their 4,000 bp flanking sequences (...) as an IStron-like***
***intron"; this section seems particularly convoluted and may need rewriting***

We apologize that this section was unclear. We have rewritten this part to improve clarity by
simplifying the description of the analytical workflow (Page 19, Lines 413-424).

***475 - "Detection of Homing Endonucleases and Reverse Transcriptases"; I'd clarify why***
***ORFs were used instead of genes (it makes sense but is not explained in the paper).***

The use of the term “gene” in the main text and “ORF” in the Methods section may have caused
confusion. We used the term ORF when analyzing coding regions present within intron
sequences. We predicted ORFs within intron regions and screened them for conserved domains
(homing endonucleases or reverse transcriptases), following established practice (Tsurumaki et
al., 2024, J. Bacteriol). We revised the manuscript to clarify this rationale (Page 22, Lines 480–
484).

***487 - "Construction of Phylogenetic Tree for Group I introns"; Are you sure MAFFT***
***method is appropriate for divergent non-coding RNAs that often display structural rather***
***than sequence similarity?***

Thank you for the valuable comments. Given that group I introns are non-coding RNAs whose
secondary structures are often more conserved than their primary sequences, we reanalyzed the
data using a structure-based alignment approach. Specifically, we replaced the MAFFT-based
sequence alignment with an alignment generated by cmalign, which incorporates covariance
models and structural information.

To focus on reliably aligned regions, we trimmed the alignment by retaining only positions
present in at least 90% of the sequences. Phylogenetic trees were then reconstructed using IQ-
TREE based on this cmalign-derived alignment. The resulting tree showed an overall topology
similar to that obtained using MAFFT, particularly with respect to the broad separation between
the IA and IC subgroups. Within the IA subgroup, which is the main focus of this study, some
differences in branching patterns were observed; however, the overall trends were largely
consistent with those inferred from the MAFFT-based analysis. Based on these results, we
replaced the original phylogenetic tree with the one constructed from the cmalign-based
alignment and revised the manuscript accordingly (Page 23, Lines 492-497).

Re: mSystems01536-25R1 (Group I Introns in tRNA Genes of Patescibacteria)

Dear Dr. Shino Suzuki:

Your manuscript has been accepted, and I am forwarding it to the ASM production staff for publication. Your paper will first be checked to make sure all elements meet the technical requirements. ASM staff will contact you if anything needs to be revised before copyediting and production can begin. Otherwise, you will be notified when your proofs are ready to be viewed.

Sincerely,
Nicola Vitulo
Editor
mSystems